# POLICY ARCHITECTURES FOR COMPOSITIONAL GENERALIZATION IN CONTROL

## ABSTRACT

Several tasks in control, robotics, and planning can be specified through desired goal configurations for entities in the environment. Learning goal-conditioned policies is a natural paradigm to solve such tasks. However, learning and generalizing on complex tasks can be challenging due to variations in number of entities or compositions of goals. To address this challenge, we introduce the Entity-Factored Markov Decision Process (EFMDP), a formal framework for modeling the entity-based compositional structure in control tasks. Geometrical properties of the EFMDP framework provide theoretical motivation for policy architecture design, particularly Deep Sets and popular relational mechanisms such as graphs and self attention. These structured policy architectures are flexible and can be trained end-to-end with standard reinforcement and imitation learning algorithms. We study and compare the learning and generalization properties of these architectures on a suite of simulated robot manipulation tasks, finding that they achieve significantly higher success rates with less data compared to standard multilayer perceptrons. Structured policies also enable broader and more compositional generalization, producing policies that **extrapolate** to different numbers of entities than seen in training, and **stitch** together (i.e. compose) learned skills in novel ways. Video results can be found at https://sites.google.com/view/comp-gen-anon.

## 1 INTRODUCTION

Goal specification is a powerful abstraction for training and deploying AI agents (Kaelbling, 1993) For instance, object reconfiguration (Batra et al., 2020) tasks, like loading plates in a dishwasher or arranging pieces on a chess board, can be described through spatial and semantic goals for various objects. In addition, the goal for a scene can be described through compositions of goals for individual *entities* in it. Through this work, we introduce a new framework for modeling tasks with such **entity-centric** compositional structure, applicable to domains like robotic manipulation, multi-agent systems, and strategic game-playing. Subsequently, we study policy architectures that can utilize structural properties unique to our framework for goal-conditioned reinforcement and imitation learning. Through experiments in simulated robot manipulation tasks, we find that our policy architectures exhibit significantly improved learning efficiency and generalization performance compared to standard multi-layer perceptrons

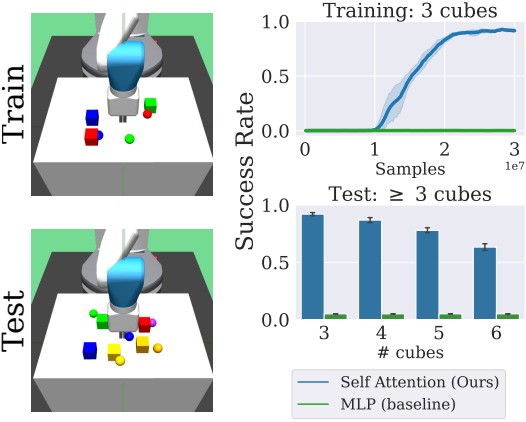

*Figure 1.* A family of tasks where the agent is trained to re-arrange three cubes (top-left), but tested zero-shot to re-arrange more cubes (bottom-left). RL with standard MLPs fails to even learn the 3-cube task. Our self-attention policy, on the other hand, successfully learns and extrapolates.

(MLPs), as previewed in Figure 1. More importantly, our architectures are capable of learning near-optimal policies in complex table top manipulation tasks where MLP baselines completely fail.

Consider the motivating task of arranging pieces on a chess board using a robot arm. A naive specification would provide goal locations for all 32 pieces simultaneously. However, we can

immediately recognize that the task is a composition of 32 sub-goals involving the rearrangement of individual pieces. This understanding of compositional structure can allow us to focus on one object at a time, dramatically reducing the size of effective state space and help combat the curse of dimensionality that plagues RL (Sutton and Barto, 1998; Bertsekas and Tsitsiklis, 1996). Moreover, such a compositional understanding would make an agent invariant to the number of objects, enabling generalization to fewer or more objects. Most importantly, it can enable reusing shared skills like pick-and-place, enhancing the learning efficiency. We finally note that a successful policy cannot completely decouple the sub-tasks. For example, if a piece must be moved to a square currently occupied by another piece, the piece in the destination square must be moved first.

The generic Markov Decision Process (MDP) framework as well as policy architectures based on MLPs lack the aforementioned compositional properties. To overcome this limitation, we turn to the general field of "geometric deep learning" (Bronstein et al., 2021) which is concerned with the study of structures, symmetries, and invariances exhibited by function classes. We first introduce the Entity-Factored MDP (EFMDP), a subclass of the generic MDP, as a formal model for decision making in environments with multiple entities (e.g. objects). We then characterize the geometric properties of EFDMP relative to the generic MDP. We subsequently show how set-based invariant architectures like Deep Sets (Zaheer et al., 2017) and relational architectures like Self-Attention (Vaswani et al., 2017) and Graph Convolution (Kipf and Welling, 2016) are well suited to leverage the geometric properties of the EFMDP. Through experiments, we demonstrate that policies and critics parameterized by these architectures can be trained to solve complex tasks using standard RL and IL algorithms, without assuming access to any options or action primitives.

**Our Contributions.** This paper is organized into sections that present our three main contributions:

1. We develop the Entity-Factored MDP (EFMDP) framework, a formal model for decision making in tasks comprising of multiple entities (e.g. objects), and characterize its geometric properties.
2. We show how policies and critics parameterized by set-based invariance models (e.g. Deep Sets) or relational models (e.g. Self-Attention and Graph Convolution) can leverage the geometric properties of the EFMDP.
3. We empirically evaluate these structured architectures on a suite of simulated robot manipulation tasks (Figure 4), and find that they generalize more broadly while also learning more efficiently. Compared to MLPs, structured policies improve success rates by more than $50\times$ on **extrapolation** tests which vary the numbers of entities in the environment, and by $10\times$ on **stitching** tests that require recombining learned skills in novel ways to solve new unsteen tasks.

## 2    PROBLEM FORMULATION AND ARCHITECTURES

In this section, we first formalize our problem setup by introducing the entity-factored MDP (EFMDP). This setting is capable of modeling many applications including table-top manipulation, scene reconfiguration, and muti-agent learning. Subsequently, we also introduce policy architectures that can enable efficient learning and generalization by utilizing the EFMDP's unique structural properties.

### 2.1    PROBLEM SETUP

We study a learning paradigm where the agent can interact with many entities in an environment. The task for the agent is specified in the form of goals for some subset of entities (including the agent). We formalize this learning setup with the Entity-Factored Markov Decision Process (EFMDP).

**Definition 1** (Entity-Factored MDP). *An EFMDP with $N$ entities is described through the tuple: $\mathcal{M}^E := \langle \mathcal{U}, \mathcal{E}, \mathfrak{g}, \mathcal{A}, \mathbb{P}, \mathcal{R}, \gamma \rangle$. Here $\mathcal{U}$ and $\mathcal{E}$ are the agent and entity state spaces, $\mathfrak{g}$ is the subgoal space and $\mathcal{A}$ is the agent's action space. The overall state space $\mathcal{S} := \mathcal{U} \times \mathcal{E}^N$ has elements $\mathbf{s} = (u, e_1, \cdots, e_N)$ and the overall goal space $\mathcal{G} := \mathfrak{g}^N$ has elements $\mathbf{g} = (g_1, \ldots, g_N)$. The reward and dynamics are described by:*

$$\mathcal{R}(\mathbf{s}, \mathbf{g}) := \mathcal{R}\left(\{\tilde{r}(e_i, g_i, u)\}_{i=1}^N\right) \tag{1}$$

$$\mathbb{P}(\mathbf{s}'|\mathbf{s}, \mathbf{a}) := \mathbb{P}\left(\left(u', \{e_i'\}_{i=1}^N\right) \mid \left(u, \{e_i\}_{i=1}^N\right), \mathbf{a}\right) \tag{2}$$

*for $\mathbf{s}, \mathbf{s}' \in \mathcal{S}$, $\mathbf{a} \in \mathcal{A}$, and $\mathbf{g} \in \mathcal{G}$.*

The EFMDP is a goal-conditioned MDP (Kaelbling, 1993; Schaul et al., 2015) with additional structure. Each entity is associated with a specific reward $\tilde{r}_i = \tilde{r}(e_i, g_i, u)$, which are aggregated together to reward the agent. The aggregation can follow various rules like requiring "all" entity subgoals to be satisfied or "any" entity subgoal be satisfied. We also note that the EFMDP does not force the entities to be exchangeable or indistinguishable, since the entity state space may contain identifying properties distinguishing each entity. The ultimate objective for the learning agent in is to learn a policy $\pi^\star : \mathcal{S} \times \mathcal{G} \to \mathcal{A}$ that maximizes the long term rewards, given by:

$$\pi^\star := \arg\max_\pi \left\{ J(\pi) := \mathbb{E}_\pi \left[ \sum_{t=0}^\infty \gamma^t \mathcal{R}(\mathbf{s}_t, \mathbf{g}) \right] \right\}. \tag{3}$$

The EFMDP can model several applications including table-top manipulation, scene reconfiguration, multi-agent learning, and strategic game playing. At the same time, the EFMDP contains more structure and symmetry compared to the standard MDP model, which can enable more efficient learning and better generalization. The crucial symmetry exists in the reward and dynamics, which treat entity-subgoal pairs as unordered sets and are therefore invariant under permutations.

**Property 1** (EFMDP Permutation Symmetry). *For any permutation $\sigma \in S_N$ (the group of all permutations of $N$ items), the reward satisfies $\mathcal{R}(\sigma\mathbf{s}, \sigma\mathbf{g}) = \mathcal{R}(\mathbf{s}, \mathbf{g})$ and the transition dynamics satisfy $\mathbb{P}(\sigma\mathbf{s}'|\sigma\mathbf{s}, \mathbf{a}) = \mathbb{P}(\mathbf{s}'|\mathbf{s}, \mathbf{a})$ for any $\mathbf{s}, \mathbf{s}' \in \mathcal{S}$ and $\mathbf{a} \in \mathcal{A}$, where:*

$$\sigma\mathbf{s} := (u, e_{\sigma(1)}, \cdots, e_{\sigma(N)}) \quad \text{and} \quad \sigma\mathbf{g} := (g_{\sigma(1)}, \cdots, g_{\sigma(N)}) \tag{4}$$

This property captures the general intuition that the ordering of entity-subgoal pairs is arbitrary and not relevant to the actual environment. We also prove that any optimal policy and the optimal value function are permutation invariant.

**Proposition 1** (Policy and Value Invariance). *In any EFMDP with $N$ entities, any optimal policy $\pi^\star : \mathcal{S} \times \mathcal{G} \to \mathcal{A}$ and optimal action-value function $Q^\star : \mathcal{S} \times \mathcal{A} \times \mathcal{G} \to \mathbb{R}$ are both invariant to permutations of the entity-subgoal pairs. That is, for any $\sigma \in S_N$:*

$$\pi^\star(\sigma\mathbf{s}, \sigma\mathbf{g}) = \pi^\star(\mathbf{s}, \mathbf{g}) \ \text{and} \ Q^\star(\sigma\mathbf{s}, \mathbf{a}, \sigma\mathbf{g}) = Q^\star(\mathbf{s}, \mathbf{a}, \mathbf{g})$$

This is a direct consequence of the permutation symmetry in reward and dynamics; we provide a proof in Appendix A. Note that Proposition 1 only talks about the optimal policy and value function, the permulation symmetry does not hold for every policy. In fact, the permutation symmetry does not hold for most commonly used architectures like MLPs. In the next subsections, we use Proposition 1 to guide architecture design in reinforcement and imitation learning on EFMDPs. We show that certain "entity-centric" model classes achieve invariance for *every* policy and value in the class, readily utilizing the structure and symettries afforded by the EFMDP.

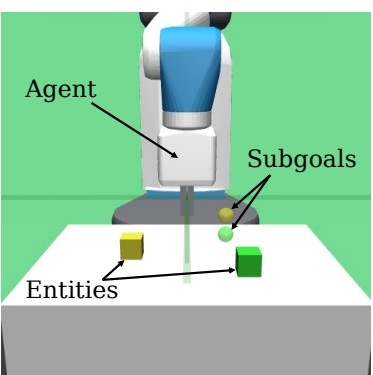

*Figure 2.* In an EFMDP, an agent interacts with entities that have corresponding subgoals. This framework can model rearrangement, strategic game playing, and multi-agent systems. In this "push and stack" example, the agent must move the green cube to its subgoal, indicated by the green sphere, and then stack the yellow cube on top of the green cube.

## 2.2 Multilayer Perceptrons (MLPs)

Standard RL and IL approaches assume they are solving a generic MDP, and do not use any additional structure. The generic approach is thus to parameterize the learned policy by an MLP, which takes a fixed size input vector and applies alternating layers of affine transforms and pointwise nonlinearities to produce a fixed size output vector. To implement $\pi(\mathbf{s}, \mathbf{g})$ with an MLP we arrange the contents of $(\mathbf{s}, \mathbf{g})$ into a single long vector using concatenation:

$$\text{vec}(\mathbf{s}, \mathbf{g}) := \text{Concatenate}(\underbrace{u, e_1, \cdots, e_N}_{=\mathbf{s}}, \underbrace{g_1, \cdots, g_N}_{=\mathbf{g}}) \tag{5}$$

Denoting the action of the MLP as a vector-to-vector function $\text{MLP}(\cdot)$, our policy is defined $\pi(\mathbf{s}, \mathbf{g}) := \text{MLP}(\text{vec}(\mathbf{s}, \mathbf{g}))$. Since MLPs expect input vectors of a fixed dimension, testing on tasks with more entities requires zero padding the inputs during training to ensure consistent input dimensionality across all tasks.

*Figure 3.* Visualizations of implementing an entity-based goal conditioned policy using either Deep Sets (left) or Self Attention (right). The policy $\pi : (\mathbf{s}, \mathbf{g}) \mapsto \mathbf{a}$ receives state $\mathbf{s} = (u, e_1, \cdots, e_N)$ containing agent state $u$ and entity states $e_i$. The goal $(g_1, \cdots, g_N)$ contains subgoals for each entity. Both policies arrange the input into $N$ vectors $y_i = (u, e_i, g_i)$, one per entity. The Deep Set policy processes each $y_i$ independently with MLP $\phi(\cdot)$, aggregates the outputs, and maps the result to an action using MLP $\rho(\cdot)$. The self attention encoder $\mathrm{SA}(\cdot)$ produces output $z_1, \cdots, z_N$ and uses self-attention to model interactions between the entities/subgoals. The $z_i$ are mapped to an action by summation and an MLP $\rho(\cdot)$.

## 2.3 DEEP SETS

The MLP policy represents a "black-box" approach to generic MDPs that fails to guarantee permutation invariance (Prop. 1). As a result, MLPs might require significant amount of data to learn the necessary permutation invariance. In contrast, the Deep Sets (Zaheer et al., 2017) (DS) architecture can guarantee permutation invariance of subgoal-entity pairs by construction. Given a set of vectors $x = \{x_1, \cdots, x_N\}$, it constructs a model of the form:

$$\mathrm{DS}(x) := \rho\left(\sum_i \phi(x_i)\right), \qquad (6)$$

where $\rho$ and $\phi$ are themselves typically MLPs. $\mathrm{DS}(\cdot)$ is invariant to ordering of $\{x_i\}$, since $\sum_i(\cdot)$ is agnostic to the ordering of elements. More surprisingly, Zaheer et al. (2017) showed that deep sets can represent *any* permutation invariant function of $x$, given that $\rho, \phi$ are sufficiently expressive. Karch et al. (2020) introduced Deep Sets for instruction following policies in a 2D environment, though to our knowledge, Deep Sets remain underexplored in more complex environments.

We now present a simple but general approach for implementing invariant policies using Deep Sets for any EFMDP. For this, we arrange the entity-subgoal pairs as a set $\{(e_1, g_1), \cdots, (e, g_N)\}$. We also include the "global" or "shared" agent state $u$ to every entity-subgoal pair. The Deep Set then produces produces an action from this set:

$$\pi(\mathbf{s}, \mathbf{g}) := \mathrm{DS}\left(\{y_i\}_{i=1}^N\right), \qquad (7)$$

$$y_i := \mathrm{Concatenate}(\underbrace{u, e_i,}_{\in \mathbf{s}} \underbrace{g_i}_{\in \mathbf{g}}). \qquad (8)$$

Figure 3 (left) visualizes how the input is arranged and processed by by the Deep Sets policy.

## 2.4 RELATIONAL MECHANISMS: GRAPHS AND SELF ATTENTION

Although Deep Sets can represent any invariant policy in theory, its design aggregates representations for all the entities through a single summation and then requires the MLP $\rho$ to handle any interactions between them. For tasks involving complex entity-entity interactions, we might desire a stronger *relational* inductive bias. Recent relational RL (Džeroski et al., 2001) approaches often model the state as a graph and use graph neural networks (GNNs) (Gori et al., 2005; Scarselli et al., 2008) to implement policies or dynamics models. As GNNs are invariant to permutations of their nodes (Bronstein et al., 2021), they can also satisfy our EFMDP invariance condition Prop. 1 if we construct the policy input as a graph of entity-subgoal pairs.

We implement and evaluate two simple relational policy architectures based on (1) graph convolutional networks (GCN) (Kipf and Welling, 2016) and (2) self attention (SA) (Vaswani et al., 2017). Although originally developed for sequence processing applications, previous relational RL architectures have already used self attention as a graph message-passing mechanism (Zambaldi et al., 2018; Li et al., 2020). To use either GNN-style architecture as a policy in a general EFMDP, we consider the input $\mathbf{s}$ as a complete graph where each node corresponds to an entity and the corresponding node features are $\{y_1, \cdots, y_N\}$, where vector $y_i$ is defined in Eq. 8. The GNN component consists of either multiple GCN layers or multiple self attention layers and transforms the input graph into an output graph with node features $\{z_1, \cdots, z_N\}$ that now capture relationships between the nodes.

Finally, the policy pools the $z_i$'s together by summation and project the result to an action $\mathbf{a} \in \mathcal{A}$ using a small MLP $\rho(\cdot)$. Figure 3 (right) illustrates the self attention policy design. In addition to satisfying permutation invariance, the relational policies use either graph convolution (GCN) or self attention (SA) to produce intermediate representations $z_i$ that include interactions between the inputs, which can be a stronger inductive bias on complex tasks.

# 3 EXPERIMENTS AND EVALUATION

In this section, we aim to study the following questions through our experimental evaluation.

1. How efficiently do structured policies **learn** a given entity-centric task?
2. Can the structured policies **extrapolate** to more or fewer entities?
3. Can the structured policies solve tasks containing novel *combinations* of subtasks, by **stitching** together (i.e. composing) learned skills?

Extrapolation and stitching are particularly interesting as they require generalization to novel tasks with no additional training. This is particularly useful when deploying agents in real world settings with enormous task diversity. Press et al. (2021) showed that self attention can achieve interesting sequence-length extrapolation behaviors in natural language processing tasks, which suggests that these architecture classes may also display interesting generalization capabilities in control tasks.

**Environment Description.** We seek to answer our experimental questions in a suite of simulated robotic manipulation environments, where the policy provides low level continuous actions to control a Fetch robot and interact with any number of cubes and switches. There are three subtasks: to *push* a cube to a desired location on the table, to flip a *switch* to a specified setting, or to *stack* one cube on top of another. The higher level tasks can involve multiple cubes or switches and compose many subtasks together, as shown in Figure 4. These environments fit naturally into the EFMDP framework: the robot is the agent, the cubes and switches are entities, and the goal specifies desired cube locations or switch settings.

We organize the environments into **families** to test learning and generalization. Environments in the *N-Push* family require re-arranging $N$ cubes by pushing each one to its corresponding subgoal. The *N-Switch* family requires flipping each of $N$ switches to its specified setting, and the *N-Switch + N-Push* family involves re-arranging $N$ cubes *and* flipping $N$ switches. We test extrapolation by varying $N$ within a family at test time, which changes the number of entities: for example we train a policy in *3-Switch* and evaluate it in *6-Switch*. As another example, we test stitching by training a single policy on *2-Switch* and *2-Push*, then evaluate it on *2-Switch + 2-Push* which requires combining the switch and pushing skills together in a single trajectory. Note that entity-entity collisions are disabled in non-stacking tasks for the main results, but Appendix D.2 repeats the *N-Push* experiments with collisions enabled. Appendix B gives a full description of our environments.

**Baselines and Comparisons.** Our main comparisons are with: (a) a baseline MLP that models the task as a regular MDP (Sec. 2.2), and (b) an "oracle" that manually coordinates solving one subtask at a time. We construct subpolicies for the oracle by training one policy on each distinct subtask (pushing, flipping switches, and stacking). The oracle chooses an initial entity and subgoal arbitrarily, and uses the corresponding subpolicy until that subtask is solved. The oracle then selects the appropriate

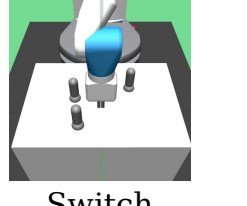
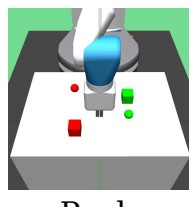
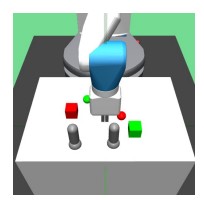
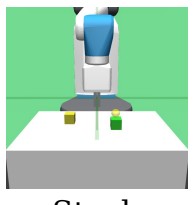

| Switch | Push | Switch+Push | Stack |
|---|---|---|---|

*Figure 4.* Illustrations of the robot manipulation environments we study. They consist of subtasks such as pushing a cube to its (spherical) target, flipping a switch to a specified position, and stacking one cube on top of another. The overall task can involve multiple entities and subtasks as well as their combinations (pushing cubes and flipping switches).

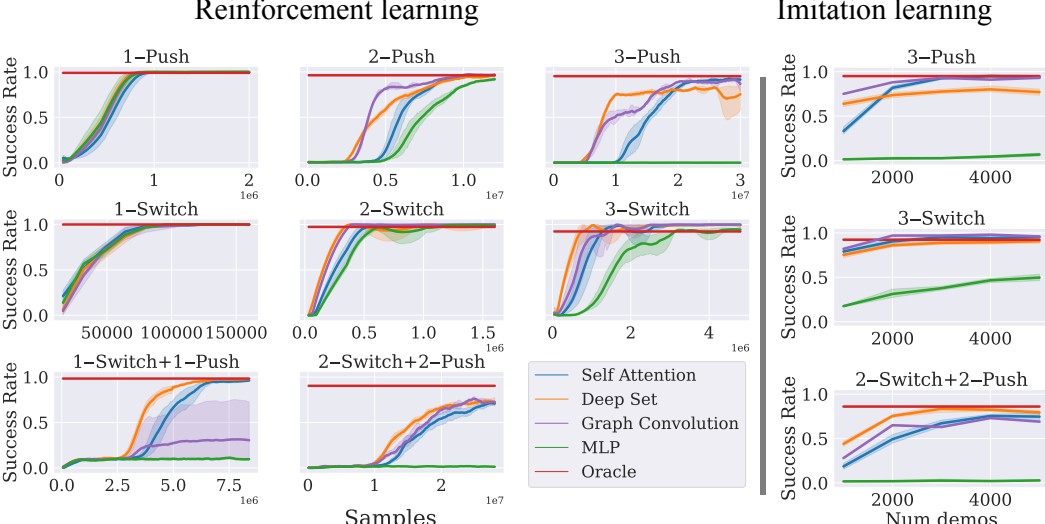

*Figure 5.* Training on environments of varying complexity using either reinforcement or imitation learning. Each row corresponds to a single environment family (*N-Push*, *N-Switch*, and *N-Switch + N-Push*), where environments with larger $N$ contain more entities and are more complex. For RL (left), each plot is a training curve of success rate vs the number of steps taken in the environment. RL with standard MLPs can solve the simpler tasks such as *1-Push*, but structured policies (Self Attention, Deep Set, and Graph Convolution) are superior on the more complex environments. For IL (right), we show success rates of behavior cloning against number of expert demonstrations in the dataset. The structured policies far outperform the MLP even when given less data. Shaded regions indicate 95% CIs over 5 seeds.

subpolicy for the next entity-subgoal pair and continues until the entire task is complete. The oracle is **not** guaranteed to achieve a 100% success rate since it does not consider entity-entity interactions. An example failure mode is pushing one cube into position but knocking another one off the table while doing so. Still, as the oracle represents a hand-crafted hierarchical approach using an entity-based task decomposition, we will compare the RL and IL agents' performance against the oracle in the following experiments.

### 3.1 EFFICIENCY OF LEARNING

To evaluate the learning efficiency of different architectures, we consider the *N-Switch*, *N-Push*, and *N-Switch + N-Push* environment families. We try $N = 1, 2, 3$ for the first two families and $N = 1, 2$ for the latter, with larger $N$ corresponding to more entities and more complex tasks within a family. **Evaluation criteria:** An episode in the environment is considered successful only if all the sub-goals in the environment are achieved.

We separately train policies on each environment in each family, using either RL or IL approaches. For RL training we use DDPG (Lillicrap et al., 2015) with Hindsight Experience Replay (HER) (Andrychowicz et al., 2017), where we use the same architecture (either MLP, Deep Set, or Self Attention) to implement **both** the policy and critic. For IL we use behavior cloning to train policies to fit a dataset of expert trajectories using mean-squared error loss. For each environment, we use a trained RL agent to generate the corresponding expert trajectory datasets. See Appendix C for full RL and IL training details. Additionally, Appendix D.3 contains further experimental comparisons of RL learning efficiency when training with Proximal Policy Optimization (Schulman et al., 2017) instead of DDPG, which show that the following RL results are not broadly sensitive to the particular choice of algorithm.

**RL results.** Figure 5 (left) shows RL training curves as a function of environment samples. In the simpler *1-Switch* and *1-Push* environments, all methods learn to solve the task fairly quickly. Once there is more than one entity, however, the structured policies learn faster than the MLP. In harder environments like *3-Push* or *N-Switch + N-Push*, the MLP fails to achieve a non-trivial success rate. Both Deep Set and Self Attention match Oracle performance in all environments except *2-Switch + 2-Push*. Graph Convolution additionally struggles in *1-Switch + 1-Push*, but still outperforms MLP.

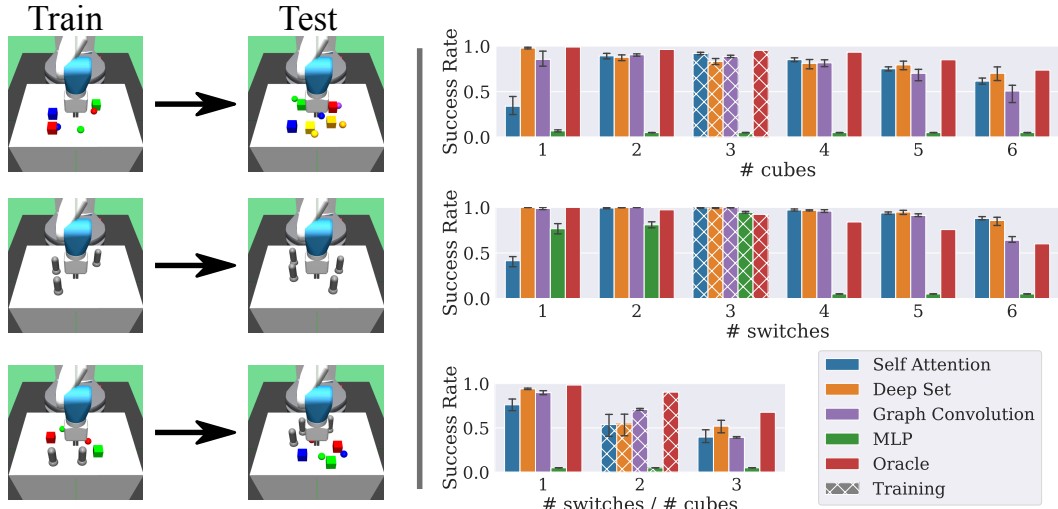

*Figure 6.* Extrapolation capabilities of RL-trained policies with different architectures. Each row depicts an environment family with a varying number of entities. Policies are trained on a *single* environment from each family before being tested on all the others, with no additional training. Bar charts show success rates in each environment, with the hatched bars corresponding to training environments. The structured policies (Self Attention, Deep Set, and Graph Convolution) extrapolate beyond the training environment to solve tasks with more or fewer entities than seen in training, while MLP policies struggle on more complex testing environments. Error bars are 95% CIs on 5 seeds.

Although they achieve similar asymptotic performance on most tasks, the Deep Set policy tends to learns faster than the others, possibly because it is simpler and has fewer parameters.

**IL Results.** The imitation learning results appear in Figure 5 (right), where the x-axis now indicates the size of the training dataset used for behavior cloning. Similar to the RL setting, we see that the structured policies learn far more efficiently than the MLP in all environments. For example, in *3-Push* with 5000 demonstrations, the MLP's success rate is still nearly zero while the Self Attention policy has a nearly 100% success rate.

**Conclusions.** MLP policies struggle to learn complex tasks with many entities with both RL and IL, likely due to the lack of entity-centric processing that the structured policies employ. The Deep Set policy typically learns faster than the others in RL, and matches or outperforms Self Attention in IL with 1000 trajectories. Although the asymptotic performance of the entity-centric methods is typically similar, the relational methods are superior to Deep Set on *3-Push* for both RL and IL. *3-Push* is one of the more difficult tasks, and relational policies may benefit from greater relational expressivity through its self attention mechanisms. Overall, this experiment suggests that architectures that utilize the structure and invariances in EFMDPs learn substantially faster than black box architectures.

## 3.2 ZERO-SHOT EXTRAPOLATION CAPABILITIES

To test whether trained policies can extrapolate and solve test tasks containing more or fewer entities than seen in training, we use the *N-Switch*, *N-Push*, and *N-Switch + N-Push* environment families. For *N-Push* and *N-Switch* we train a policy with RL on $N = 3$ and test with $N \in \{1, \ldots, 6\}$. For *N-Switch + N-Push* we train a policy with RL on $N = 2$ and test on $N \in \{1, \ldots, 3\}$. For testing, we use the RL agent checkpoint with the highest success rate in its training environment.

**Results and Observations** Figure 6 shows the test performance of these policies on each environment family as the number of entities $N$ varies. The MLP only successfully learns the training task in the *N-Switch* environments, and it generalizes decently to *fewer* than 3 switches, but fails completely in environments with *more* than 3 switches.

In contrast, the structured policies generalize well and achieve zero-shot success rates comparable to or exceeding the Oracle in most test environments. Notably, these policies well exceed oracle performance on *6-Switch* despite training in *3-Switch*. Interestingly, Self Attention policies fare poorly on single-entity test environments, perhaps because the self attention mechanism relies critically on

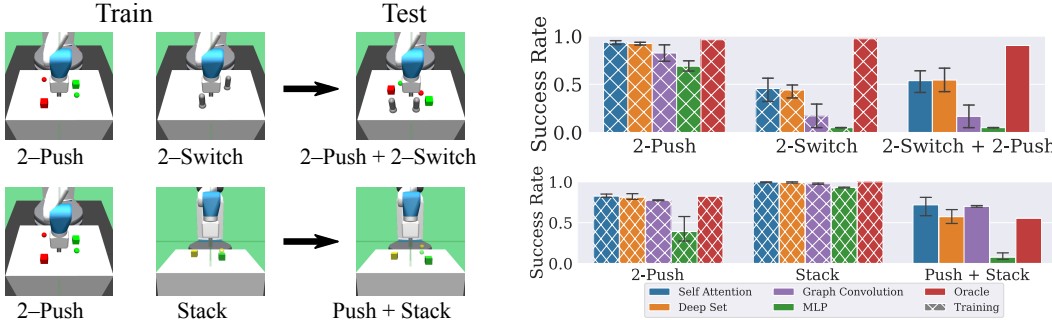

*Figure 7.* **Left**: train/test setups that require solving test tasks by stitching together training skills, with no additional data. Top: train on *2-Switch* and *2-Push*, test on *2-Switch+2-Push*. Bottom: trained on *2-Push* and *Stack*, test on *Push + Stack*. **Right**: average success rates by architecture. Deep Set and Self Attention policies are moderately successful at solving the test tasks, and are comparable to the Oracle in *Push + Stack*. The MLP fails to achieve nontrivial success rates on both test environments. Error bars indicate 95% CIs over 5 seeds.

interactions between more than one entity during training. Despite its relative simplicity, the Deep Set architecture extrapolates as well as or better than the relational architectures in most environments. A crucial exception is in *3-Push* with cube-cube collisions enabled (Appendix D.2). There, modeling entity-entity interactions is especially crucial and a relational method like Self Attention is largely superior. Overall, we find that geometric architectures can perform very effective extrapolation.

### 3.3 ZERO-SHOT STITCHING TO SOLVE NOVEL TASKS

When evaluating policies for stitching behavior, we use test tasks that combine subtasks from training in novel ways. In our first setting, we train a policy on *2-Push* and *2-Switch*, and then test this policy on *2-Switch + 2-Push*, which requires both pushing cubes *and* flipping switches. In our second setting, we train a single policy on *2-Push* and *Stack*, which requires stacking one cube on top of another. The test environment is *Push + Stack*, which requires pushing one cube into position and then stacking the other block on top. This setting is especially difficult because it requires zero-shot stitching of skills in a particular order (push, *then* stack). Figure 7 (left) shows the train-test task relationships we use to test stitching.

**Results and Observations** Since this experiment requires training a single policy on multiple training tasks, during each episode we choose one of the training tasks uniformly at random. Figure 7 (right) shows that the MLP policy fails to jointly learn the training tasks in the first setting, leading to poor performance in *2-Switch + 2-Push*. However, the MLP averages above a 35% success rate on both training tasks in the second setting, but still only manages a 5% success rate on *Push + Stack*. This suggests that even when MLP policies are capable of learning the training tasks, they are unable to combine them to solve new ones.

The geometric architectures show substantially better (but not oracle-level) stitching capabilities compared to the MLP. Graph Convolution struggles with the switch component of *2-Switch + 2-Push*, but still outperforms the MLP. It is particularly surprising that the two relational architectures (Self Attention and Graph Convolution) achieve > 60% zero-shot success rate on *Push + Stack*, which requires understanding that the push and stack subtasks must be executed in a specific order. Poor performance in *2-Switch + 2-Push* is again due to difficulties in training one policy on two different tasks, suggesting that better joint training could further improve stitching performance.

## 4 RELATED WORK

**Compositionality and Hierarchy.** Hierarchical approaches to solving long-horizon tasks explicitly maintain or learn subpolicies corresponding to useful skills, which can then be coordinated by a high-level policy (Dayan and Hinton, 1993; Parr and Russell, 1998; Dieterich, 2000). Variations of this approach include learning termination policies for each sub-policy or "option" (Bacon et al., 2017), training the high level policy to propose subgoals for goal-conditioned low level skills (Nachum et al., 2018), or even using natural language as the interface between high and low level policies (Jiang et al., 2019). Our approach also enables learning of long-horizon and compositional tasks, but simply

through architectural modifications to the policy in end-to-end learning, as opposed to explicitly learning action representations or modifying the training process.

**Entity-centric modeling.** Recent works in relational RL (Džeroski et al., 2001) have investigated graph neural networks (GNNs) (Gori et al., 2005; Scarselli et al., 2008) for handling complex multi-entity tasks, where the relational or message passing mechanism may be implemented using self attention (Zambaldi et al., 2018; Li et al., 2020) or a variety of other means (Bapst et al., 2019; Lin et al., 2022). Though we directly focus on policy (and critic) architecture for model free learning, related approaches have studied entity structured networks for dynamics models (Carvalho et al., 2021; Veerapaneni et al., 2020; Sanchez-Gonzalez et al., 2018). Further work has explored relational architectures for extracting entity-centric representations from high dimensional observations before doing control (Wilson and Hermans, 2020; Driess et al., 2022). In particular, Zadaianchuk et al. (2020) combine entity-centric representation learning with a goal conditioned RL approach that also demonstrated extrapolation, though not stitching. In settings that can be modeled as EFMDPs, our framework formally motivates using GNNs through the perspective of permutation invariance (Bronstein et al., 2021). Tang and Ha (2021) study the permutation invariance of self attention policies in particular, mainly in the context of robustness to input corruptions. But the invariance properties of our EFMDP framework also suggest considering invariant architectures *without* self attention or other relational mechanisms, in which case GNNs reduce to simpler architectures like Deep Sets (Zaheer et al., 2017). These architectures remain relatively underexplored outside of basic 2D environments (Karch et al., 2020). Our experiments evaluate *both* relational and Deep Set approaches on a suite of complex entity-centric robot tasks.

**Policy Architectures in RL.** MLPs, LSTMs, and small CNNs remain the dominant architectures in continuous control (Lillicrap et al., 2015; Schulman et al., 2017; Haarnoja et al., 2018). Sinha et al. (2020) study deeper networks for continuous control with DenseNet-style (Huang et al., 2017) connections. Recent work has also explored the use of self attention over the trajectory history rather than between entities (Chen et al., 2021; Janner et al., 2021). Other approaches leverage inductive biases about the real world, e.g. by embedding learnable dynamical systems into the policy architecture (Bahl et al., 2020).

## 5  CONCLUSION

This work introduces the EFMDP framework for the learning paradigm where an agent can interact with many entities in an environment. We explore how structural properties of EFMDPs induce a permutation symmetry in the optimal policy and value functions, motivating policy architectures that leverage symmetry: set-based invariant models (Deep Sets) and relational models (Self Attention and Graph Convolution). These policy architectures decompose goal-conditioned tasks into their constituent entities and subgoals. These architectures are flexible, do not require any manual task annotations or action primitives, and can be trained end-to-end with standard RL or IL algorithms.

We compare these architecture types with each other and standard MLPs in a suite of complex entity-centric tasks. We find that geometric architectures can: (a) **learn substantially faster** than black-box architectures like the MLP; (b) perform **zero-shot extrapolation** to new environments with more of fewer entities than observed in training; and (c) perform **zero-shot stitching** of learned behaviors to solve novel task combinations never seen in training. We find that the geometric architectures perform relatively similarly across most tasks, which can be surprising given the Deep Set's relative simplicity. Since many existing entity-centric approaches focus on graph neural networks or transformers, our results invite further investigation into simple invariant architectures like Deep Sets.

**Limitations and Future Work:** EFMDPs require entity-specific subgoals, but some tasks may instead be specified in terms of entity relations ("place this block on top of that block"). In such cases, the relational subgoals must be first converted into an equivalent entity specific form. Additionally, EFMDP's invariance properties explain why structured policies perform well, but do not distinguish between them (e.g., DS vs SA). We make this comparison empirically here, but future work could provide a more principled framework for choosing the right architecture for a given entity-centric task. We also hope to analyze the how the geometric properties of EFMDPs interact with object-centric representation learning, a vibrant area of research (Burgess et al., 2019; Kipf et al., 2019; Locatello et al., 2020; Nanbo et al., 2020) which is important to enabling compositional generalization for policies that operate on high dimensional observations like images.

**Reproducibility**: Appendix B describes the environments, and Appendix C describes the training details. We have already open-sourced the code implementing the environments and experiments, and will include the link after the anonymous reviewing phase ends.

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

## A   PERMUTATION INVARIANCE

We recall Proposition 1:

**Proposition 1** (Policy and Value Invariance). *In any EFMDP with $N$ entities, any optimal policy $\pi^\star : \mathcal{S} \times \mathcal{G} \to \mathcal{A}$ and optimal action-value function $Q^\star : \mathcal{S} \times \mathcal{A} \times \mathcal{G} \to \mathbb{R}$ are both invariant to permutations of the entity-subgoal pairs. That is, for any $\sigma \in S_N$:*

$$\pi^\star(\sigma\mathbf{s}, \sigma\mathbf{g}) = \pi^\star(\mathbf{s}, \mathbf{g}) \ \text{ and } \ Q^\star(\sigma\mathbf{s}, \mathbf{a}, \sigma\mathbf{g}) = Q^\star(\mathbf{s}, \mathbf{a}, \mathbf{g})$$

We want to show that any optimal policy $\pi^\star : \mathcal{S} \times \mathcal{G} \to \mathcal{A}$ and the optimal action-value function $Q^\star : \mathcal{S} \times \mathcal{A} \times \mathcal{G} \to \mathbb{R}$ are both permutation invariant, that is for any $\sigma \in S_N$:

$$\pi^\star(\sigma\mathbf{s}, \sigma\mathbf{g}) = \pi^\star(\mathbf{s}, \mathbf{g}) \tag{9}$$

$$Q^\star(\sigma\mathbf{s}, \mathbf{a}, \sigma\mathbf{g}) = Q^\star(\mathbf{s}, \mathbf{a}, \mathbf{g}) \tag{10}$$

Recall that in an EFMDP the reward and dynamics have permutation symmetry (Property 1):

$$\mathcal{R}(\mathbf{s}, \mathbf{a}, \mathbf{g}) = \mathcal{R}(\sigma\mathbf{s}, \mathbf{a}, \sigma\mathbf{g})$$

$$\mathbb{P}(\mathbf{s}'|\mathbf{s}, \mathbf{a}) = \mathbb{P}(\sigma\mathbf{s}'|\sigma\mathbf{s}, \mathbf{a})$$

where $\sigma\mathbf{s}$ and $\sigma\mathbf{g}$ are defined in Eq. 4. We assume for simplicity that the agent space $\mathcal{U}$ and entity space $\mathcal{E}$ are discrete, so that the state space $\mathcal{S} = \mathcal{U} \times \mathcal{E}^N$ is also discrete.

We begin with $Q^\star$, which can be obtained by value iteration, where $Q_k^\star$ denotes the $k$'th iterate. We initialize $Q_0^\star \equiv 0$, which is (trivially) permutation invariant. Permutation invariance is then preserved during each step of value iteration $Q_k^\star \mapsto Q_{k+1}^\star$:

$$Q_{k+1}^\star(\sigma\mathbf{s}, \mathbf{a}, \sigma\mathbf{g}) = \mathcal{R}(\sigma\mathbf{s}, \mathbf{a}, \sigma\mathbf{g}) + \gamma \max_{\mathbf{a}'} \sum_{\mathbf{s}' \in \mathcal{S}} \mathbb{P}(\mathbf{s}'|\sigma\mathbf{s}, \mathbf{a}) Q_k^\star(\mathbf{s}', \mathbf{a}') \tag{11}$$

$$= \mathcal{R}(\mathbf{s}, \mathbf{a}, \mathbf{g}) + \gamma \max_{\mathbf{a}'} \sum_{\mathbf{s}' \in \mathcal{S}} \mathbb{P}(\sigma^{-1}\mathbf{s}'|\mathbf{s}, \mathbf{a}) Q_k^\star(\sigma^{-1}\mathbf{s}', \mathbf{a}') \tag{12}$$

$$= \mathcal{R}(\mathbf{s}, \mathbf{a}, \mathbf{g}) + \gamma \max_{\mathbf{a}'} \sum_{\mathbf{s}' \in \mathcal{S}} \mathbb{P}(\mathbf{s}'|\mathbf{s}, \mathbf{a}) Q_k^\star(\mathbf{s}', \mathbf{a}') \tag{13}$$

$$= Q_{k+1}^\star(\mathbf{s}, \mathbf{a}, \mathbf{g}) \tag{14}$$

Hence $Q_k^\star$ is permutation invariant for all $k = 0, 1, \cdots$, with $Q_k^\star \xrightarrow[k \to \infty]{} Q^\star$. Line 12 follows from the permutation invariance of the reward, transition probability, and the previous iterate $Q_k^\star$. Line 13 uses the fact that summing over $\sigma^{-1}\mathbf{s}'$ for all $\mathbf{s}' \in \mathcal{S}$ is the same as simply summing over all states $\mathbf{s}' \in \mathcal{S}$. This can be seen more explicitly by expanding a sum over arbitrary function $f(\cdot)$:

$$\sum_{\mathbf{s} \in \mathcal{S}} f(\sigma^{-1}\mathbf{s}) = \sum_{u \in \mathcal{U}} \sum_{e_1 \in \mathcal{E}} \cdots \sum_{e_N \in \mathcal{E}} f(u, e_{\sigma^{-1}(1)}, \cdots, e_{\sigma^{-1}(N)}) = \sum_{\mathbf{s} \in \mathcal{S}} f(\mathbf{s})$$

The permutation invariance of $Q^\star$ leads to the permutation invariance of $\pi^\star$:

$$\pi^\star(\sigma\mathbf{s}, \sigma\mathbf{g}) = \arg\max_{\mathbf{a}} Q^\star(\sigma\mathbf{s}, \mathbf{a}, \sigma\mathbf{g}) = \arg\max_{\mathbf{a}} Q^\star(\mathbf{s}, \mathbf{a}, \mathbf{g}) = \pi^\star(\mathbf{s}, \mathbf{g})$$

## B    ENVIRONMENTS

Our environments are modified from OpenAI Gym's Fetch environments (Brockman et al., 2016) (MIT license), with our stacking environment in particular being modified from the Fetch stacking environments of Lanier (2019). They have a 4D continuous action space with 3 values for end effector displacement and 1 value for controlling the distance between the gripper fingers. The final action is disabled when the neither the training or test tasks involve stacking, since gripping is not required for block pushing or switch flipping. Input actions are scaled and bounded to be between $[-1, 1]$. We set the environment episode length based on the number of entities and subtasks involved. Each switch added 20 timesteps, and each cube pushing or stacking task added 50 timesteps. For example, *2-Switch + 2-Push* had a max episode length of $2 \times 50 + 2 \times 20 = 140$ timesteps.

For non-stacking settings such as *N-Push* and *N-Switch + N-Push*, we disable cube-cube collision physics to make training easier for all methods. Note that subgoals may still interfere with each other since the gripper can interact with all cubes, so the agent may accidentally knock one cube away when manipulating another one. We repeat the extrapolation experiments for *N-Push with collisions* in Appendix D.2.

**State and goals.** The agent state describe the robot's end effector position and velocity the gripper finger's positions and velocities. The entity state for cubes include the cube's pose and velocity, and for switches include the switch setting $\theta \in [-0.7, 0.7]$ and the position of the switch base on the table. The switch entity state is padded with zeros to match the shape of the cube entity state, and all entity states include an extra bit to distinguish cubes from switches. Subgoals specify a target position for cubes and a target setting $\theta^\star \in \{-0.7, 0.7\}$ for switches.

**Reward.** The dense reward is defined as the average distance between each entity and its desired state as specified by the subgoal. For cubes, this is the L2 distance between current and desired position. For switches, this is $|\theta - \theta^\star|$, where $\theta$ is the current angle of the switch and $\theta^\star$ is the desired setting. The sparse reward is $0$ if all entities are within a threshold distance of their subgoals, and $-1$ otherwise.

*Table 1.* General shared RL hyperparameters

| Hyperparameter | Value |
|---|---|
| Discount $\gamma$ | 0.98 |
| Parallel envs | 16 |
| Replay buffer size | $10^6$ |
| Relabel prob | 0.8 |
| Ratio of episodes : updates | $2 : 5$ |
| Optimizer | Adam |
| Learning rate | MLP, Deep Set: 0.001
Self Attention: 0.0001 |
| Reward Scale | Sparse: 1; Dense: 5 |
| Action noise $\eta_0$ (initial) | 0.2 |
| Random action prob $\epsilon_0$ (initial) | 0.3 |

## C   TRAINING DETAILS

### C.1   REINFORCEMENT LEARNING

We train RL agents using a publicly available (MIT license) implementation[1] of DDPG (Lillicrap et al., 2015) and Hindsight Experience Replay (HER) (Andrychowicz et al., 2017). Table 1 contains the default hyperparameters shared across all experiments. Our modified implementation collects experience from 16 environments in parallel into a single replay buffer, and trains the policy and critic networks on a single GPU. We used an internal cluster to parallelize experimentation across multiple random seeds and algorithms/hyperparameters. We collect 2 episodes for every 5 gradient updates, and for HER we relabel the goals in $80\%$ of sampled minibatches (the "relabel prob"). The reward scale is simply a multipler of the collected reward used during DDPG training. For exploration we use action noise $\eta$ and random action probability $\epsilon$; the output action is:

$$\tilde{a} \sim \begin{cases} a + \mathcal{N}(0, \eta), & \text{with prob } 1 - \epsilon \\ \text{Uniform}(-1, 1), & \text{with prob } \epsilon \end{cases}$$

Table 2 shows environment specific RL hyperparameters. "Epochs" describes the total amount of RL training done, with 1 epoch corresponding to $50 \times$ PARALLEL ENVS episodes. Sparse reward is used for the simpler environments, and dense reward for the harder ones. For some environments we decay the exploration parameters $\eta, \epsilon$ by a ratio computed per-epoch. Lin(.01, 100, 150) means that $\eta, \epsilon$ are both decayed linearly from $\eta_0$ and $\epsilon_0$ to $.01 \times \eta_0$ and $.01 \times \epsilon_0$ between epochs 100 and 150. The constant exploration decay schedule maintains the initial $\eta_0, \epsilon_0$ values throughout training. The target network parameters are updated as $\theta^{\text{target}} \leftarrow (1 - \tau)\theta + \tau\theta^{\text{target}}$, where $\tau$ is the target update speed.

We use the same RL hyperparameters regardless of architecture type except that the learning rate is lower for Self Attention and the exploration decay schedule may vary. Where Table 1 lists "Fast" and "Slow" decay schedules, we sweep over both options for each architecture and use the schedule that works best. In each case, the Self Attention policy prefers the slower exploration schedule and Deep Sets prefers the faster one, while the MLP typically fails to learn with either exploration schedule on the more complex environments.

**Architectures.** The exact actor and critic architectures uses for each architecture family is shown in Table 3. Linear(256) represents an affine layer with 256 output units. ReLU activations follow every layer except the last. The final actor layer is followed by a Tanh nonlinearity, and the critic has no activation function after the final layer. $A$ represents the action space dimension, and Block(N, M, H) represents a Transformer encoder block (Vaswani et al., 2017) with embedding size $N$, feedforward dimension $M$, and $H$ heads. We disable dropout within the Transformer blocks for RL training.

---

[1]https://github.com/TianhongDai/hindsight-experience-replay

*Table 2.* Environment specific RL hyperparameters

| Environment | Reward | Epochs | Exploration decay | Target update speed $\tau$ |
|---|---|---|---|---|
| 1-Push | Sparse | 50 | Constant(1) | 0.95 |
| 2-Push | Dense | 150 | Lin(.01, 75, 125) | 0.99 |
| 3-Push | Dense | 250 | Fast: Lin(.01, 30, 80) | 0.99 |
| | | | Slow: Lin(.01, 100, 175) | |
| {1,2,3}-Switch | Sparse | {10, 50, 100} | Constant(1) | 0.95 |
| 1-Switch + 1-Push | Dense | 150 | Lin(.01, 60, 100) | 0.99 |
| 2-Switch + 2-Push | Dense | 250 | Fast: Lin(.01, 75, 150) | 0.99 |
| | | | Slow: Lin(.01, 100, 150) | |

*Table 3.* RL architectures

| Family | Actor | Critic |
|---|---|---|
| MLP | Linear(256)×3, Linear(A) | Linear(256) ×3, Linear(1) |
| Deep Set | $\phi$: Linear(256) ×3
$\rho$: Linear(A) | $\phi$: Linear(256) ×2
$\rho$: Linear(256), Linear(1) |
| Self Attention | SA: Linear(256), Block(256, 256, 4)×2
$\rho$: Linear(A) | SA: Linear(256), Block(256, 256, 4)×2
$\rho$: Linear(1) |
| Graph Convolution | Linear(256) ×3, GraphConv(256) ×2
$\rho$: Linear(A) | Linear(256) ×3, GraphConv(256) ×2
$\rho$: Linear(1) |

*Table 4.* Parameter count (3-Push)

| # parameters | Actor | Critic |
|---|---|---|
| MLP | 150,020 | 150,273 |
| Deep Set | 140,292 | 140,545 |
| Self Attention | 800,260 | 800,513 |
| Graph Convolution | 271,876 | 272,129 |

## C.2 IMITATION LEARNING

The IL dataset is generated using the best performing RL agent in that environment–we record $M \in \{1000, 2000, 3000, 4000, 5000\}$ demonstration trajectories. This creates a dataset of $M \times T$ transitions $\mathcal{D} = \{(\mathbf{s}_i, \mathbf{a}_i)\}_{i=1}^{M \times T}$ for behavior cloning. However, in practice we filter the dataset slightly by discarding the transitions corresponding to trajectories that are not successful.

We use the same policy architectures shown in Table 3 and optimize mean squared error loss over the dataset:

$$\arg \min_\pi J(\pi) := \frac{1}{|\mathcal{D}|} \sum_{(\mathbf{s},\mathbf{a}) \sim \mathcal{D}} ||\pi(\mathbf{s}) - \mathbf{a}||^2$$

We use the Adam (Kingma and Ba, 2014) optimizer with learning rate 0.001 (MLP, Deep Sets) or 0.0001 (Self Attention). Each policy is trained for $60,000$ gradient steps with a batch size of $128$.

## C.3 TRAINING AND INFERENCE SPEED

Here we consider the computational complexity of using different architecture classes (MLPs, Deep Sets, and Self Attention), as we scale the number of entities $N$. We consider the number of parameters, activation memory, and computation time (for a forward pass). For MLPs with fixed hidden layer sizes, the number of parameters and computation time increase linearly with $N$ while the memory required for activations stays fixed (due to fixed hidden layer sizes). In Deep Sets and Self Attention, the number of parameters does not depend on the number of entities $N$. The activation memory and computation time grow linearly in Deep Sets, and quadratically for the pairwise interactions of Self Attention. In practice, the number of entities $N$ is modest in all our environments (e.g., fewer than 10), but computational complexity may be relevant in more complex scenes with lots of entities.

For a more holistic real-world comparison of execution and training speed, Figure 8 shows both inference time and training time in the *N-Push* environments for $N \in \{1, 2, 3\}$. The inference time is the number of milliseconds it takes an actor do a single forward pass (using a GPU) on a single input observation. The Self Attention policy involves more complex computations and is significantly slower than Deep Set and MLP policies. The RL training time is the actual number of hours required to run the reinforcement learning algorithms of Figure 5, for each architecture. Not surprisingly, we see that *3-Push* takes significantly longer to train than *1-Push*, since it is a harder environment. For a fixed environment, however, all three architecture types are comparable in speed, with the Self Attention version being slightly slower than the others. The surprising similarity in RL training time (despite much slower inference time for the Self Attention policy) suggests that most of the RL time is spent on environment simulation rather than policy or critic execution. Hence, the difference between architectures presented in this paper has only a minor effect on reinforcement learning speeds in practice.

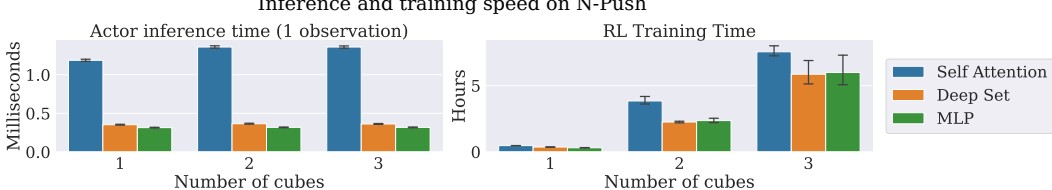

*Figure 8.* Left: the time (in milliseconds) it takes for each policy architecture to execute a single forward pass on a single input observation from the *N-Push* environments, where $N \in \{1, 2, 3\}$. The self attention policy is significantly slower, while the Deep Set and MLP policies are comparable. Right: Real world reinforcement learning times (in hours) training each policy/critic architecture on the *N-Push* environments. Although the Self Attention policy is slightly slower, all policies train at comparable speeds in the same environment. This suggests that environment simulation, not policy execution, is the dominant time consuming element.

# D FURTHER COMPARISONS

## D.1 DEEP SET ARCHITECTURE SIZE

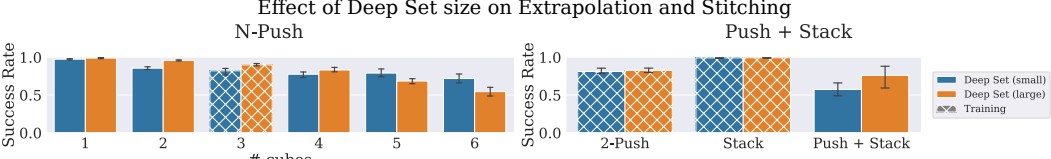

*Figure 9.* Comparison of *N-Push* extrapolation and *Push + Stack* stitching performance when using small and large variants of the Deep Set policy architecture. The small version implements $\rho$ with a 1-layer linear map, while the large version implements $\rho$ with a 2-layer MLP. For *N-Push*, the larger network achieves greater success rates in the training environment (3 cubes) but is actually worse when extrapolating to 5 or 6 cubes. On the other hand, the larger Deep Set displays superior stitching capability and achieves a higher average success rate when generalizing to *Push + Stack* from *2-Push* and *Stack*.

Recall that our Deep Set policy architecture involves two MLPs $\phi$ and $\rho$, where $\phi$ produces intermediate representations for each entity, those intermediate representations are summed, and then $\rho$ produces the final output (Eq. 6). In full generality, both $\phi$ and $\rho$ may have two or more layers with nonlinearities in between. While our $\phi$ is a 3-layer MLP, we use a *linear* $\rho$ throughout the main paper because we found that it often works comparably or better than using a larger 2-layer MLP $\rho$. Here we repeat the *N-Push* extrapolation and *Push + Stack* stitching experiments from the main paper using a 2-layer $\rho$, which we call "Deep Set (large)." The results from the main paper uses a 1-layer $\rho$ which we refer to here as "Deep Set (small)."

Figure 9 shows the results. In *N-Push*, the larger Deep Set model achieves higher training success rates in the 3-cube environment, but has worse extrapolation success rates for large numbers of cubes. For example, the smaller Deep Set model is significantly better at solving *6-Push*. Meanwhile, the large and small Deep Sets achieve very similar results in the pushing and stacking training environments. However, the larger Deep Set model achieves a higher success rate in the *Push + Stack* environment, indicating superior stitching capability. This suggests that simpler Deep Set architectures may be better for extrapolating to a large number of entities, but more complex architectures may be superior for solving complex tasks with a fixed number of entities.

## D.2 N-PUSH WITH CUBE-CUBE COLLISIONS

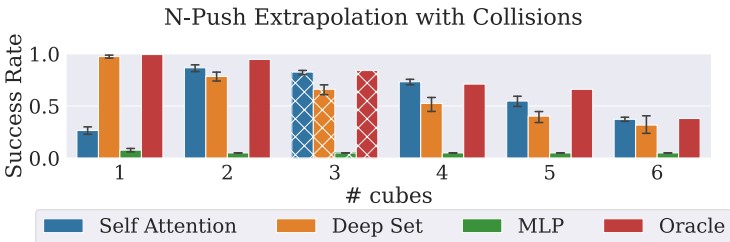

*Figure 10. N-Push* extrapolation with cube-cube collisions enabled. All methods observe some drop in performance relative to Figure 6, where *N-Push* has cube-cube collisions disabled. Self Attention tends to outperform Deep Sets when collisions enabled, likely because its relational inductive biases are better suited to handling interactions between entities that arise from collisions.

As noted in Appendix B, we disable cube-cube collisions in the *N-Push* and *N-Switch+N-Push* experiments of the main paper (of course, the stacking settings require cube-cube collisions to be enabled). Here we repeat the *N-Push* extrapolation experiments with cube-cube collisions *enabled*. Figure 10 shows the results, which are qualitatively similar to when collisions are disabled. All methods observe a decrease in success rates of about 15%, with the Self Attention method often outperforming the Deep Set policy. This is likely because *N-Push* involves more interaction between

entities once cube-cube collisions are enabled, and Self Attention's relational inductive biases are better suited for modeling these interactions.

### D.3 N-PUSH WITH PPO INSTEAD OF DDPG

Our RL learning efficiency experiments all use DDPG with Hindsight Experience Replay, which raises the question of whether the results (in particular, the inefficiency of the MLP) are specific to a particular training algorithm. To answer this, we repeated the *3-Push* RL experiment for the Deep Set and MLP architectures, but trained with Proximal Policy Optimization (PPO (Schulman et al., 2017)) instead of DDPG. We use the Stable Baselines 3 (Raffin et al., 2021) (MIT License) implementation of PPO with default hyperparameters, trained for up to $1 \times 10^6$ environment steps. Figure 11 shows that the general trend we observed while using DDPG still holds: the invariant Deep Set policy learns the task far more efficiently than the MLP. In fact, as with DDPG the MLP policy fails to learn *3-Push* at all.

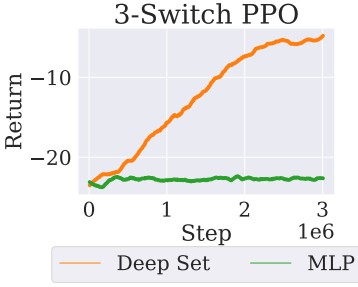

*Figure 11.* Return throughout RL training on *3-Push* using Proximal Policy Optimization (PPO) instead of DDPG.

