# OpenReview forum: "Policy Architectures for Compositional Generalization in Control"
_ICLR.cc/2023/Conference — Submitted to ICLR 2023_

### Official Review · Reviewer_KcU5 · 2022-10-21

**Confidence:** 3
**Correctness:** 2
**Technical Novelty And Significance:** 2
**Empirical Novelty And Significance:** 2
**Recommendation:** 3

**Clarity, Quality, Novelty And Reproducibility:**

Please see my strengths and weakness for clarity, quality, and novelty. The training setting is not clearly specified so reproducibility is limited.


**Strength And Weaknesses:**

Strengths:
* The paper is clearly written.
* The problem studied is interesting and important.
* Results quantitatively show the effectiveness of the proposed method.

Weaknesses:
* The proposed architectures are not novel (according to the authors, the application of these architectures on the experimented robotic tasks is novel; I don't work on this space so I am not qualified to verify this claim).
* The authors show mostly evaluation metrics without analyzing the behavior of the learned policies. How do the learned policies solve the entity tasks? Sequentially or simultaneously? Visualizing the change in the attention over the goals may offer such insights.
* I am not sure why the "oracle" is called an "oracle" rather than a rule-based baseline. The results would be interpreted much differently if the "oracle" were considered as a baseline.
* I am not convinced why the approach would work better than learning a policy separately for each entity. The authors mention that entity-entity relationships may not be considered in such an approach, but I can easily enforce some constraints to the policies through rewards or demonstrations (e.g. don't push other blocks while moving a block). The proposed method may resort to learning the "oracle" policy where it only attends to one goal at a time. Analyzing the behavior of the learned policy is important to show that the method learns a non-trivial policies. The current numerical results show that most of the time the method is on par with the "oracle" baseline, which hints that it may have learned trivial policies.
* The proposed architecture can outperform MLP because it simply has more learning capacity. The authors should show the number of parameters of the models in the comparisons.
* When training the MLP, do you permute the input entity subgoals in each episode or do you always feed the subgoals in the same order? Doesn't permuting the subgoal order make the MLP somewhat invariant to permutation?
* The problem studied, although interesting, is of limited practicality. In the real life, the decomposition into entity goal is usually inferred rather than given explicitly to the policy as in this work (e.g. "switch all light off" rather than "switch light 1 off, switch light 2 off, ..."). I understand that the authors are studying a sub-problem of this more difficult setting, but I just want to point out a limitation of the current task formulation.
* The conclusion says "These policy architectures decompose goal-conditioned tasks into their constituent entities and subgoals" but this seems like a false claim, as the goal specification already consists of entity subgoals stitched together. Figure 3 illustrating the architectures also does not show any goal decomposition.  Am I missing something?

**Summary Of The Paper:**

The paper presents an architecture that leverages the symmetry of entity-centric goal specification. The author extends the MDP framework to modeling the symmetric structure of the goal specification. The framework motivates using architectures that are agnostic to permutation of the inputs like Deep Set, Self-Attention, and Graph convolution. Experiments are conducted in two robotic tasks, showing the advantages of the proposed architectures over the MLP architecture.



**Summary Of The Review:**

The paper applies recent state-of-the-art architectures to a special class of robotic tasks. The author provides a theoretically-motivated justification but I don't find it providing significant value to the community, because the authors do not provide analyses to support the justification. I am leaning towards rejecting the paper.

=====After Rebuttal======

I have read the responses from the authors and decided to keep my current score. My main ground for rejection is that the proposed framework lacks technical depth and it does not lead to novel methods, surprising findings, or useful insights. The framework justifies using architectures that are permutation-invariant for problems that require that property, but such a choice seems to be quite natural even without considering the framework. In addition, it is unclear from the experimental results whether the data provided to the agent enforces permutation invariance and the learned policies are verified to have actually learned that property. Showing improved performance on various task conditions is not sufficient; there could be various other properties of the experimented architectures that affect their performance. More ablation studies need to be conducted to identify the cause of the improvements and to demonstrate that the actual cause coincides with the anticipated cause.

---

> ### Author Response · Authors · 2022-11-18
> **Response to Reviewer KcU5 (1/2)**
>
> Thank you for the feedback and the opportunity to answer. We believe there may be some misunderstandings in the interpretation of paper claims and results, which we hope to clarify below.
>
> **Q:** The proposed architectures are not novel
>
> **A:** Our contribution is to introduce the EFMDP framework, which provides a principled explanation for **why** invariant architectures like Deep Sets, self attention, and GNNs should work well in multi-entity control tasks. Such a formulation and understanding is not explicit in prior work. We **do not** claim to invent new architectures in this work. Experimentally, we perform a thorough and systematic evaluation to compare several architectures (generic MLPs as well as structured policies) in the same environment. We are not aware of any prior works that perform such a systematic study.
>
> ---
>
> **Q:** How do the learned policies solve the entity tasks? Sequentially or simultaneously? Visualizing the change in the attention
>
> **A**: Videos of both policy behavior and attention maps can be viewed on the [linked website](https://sites.google.com/view/comp-gen-anon). Since there is only one robot gripper, the policies tend to complete subtasks one at a time, though the behavior is fluid. We can see the policy (and attention masks) quickly switching focus between objects over the course of task completion. We have expanded the results section to include this analysis and to reference the videos on the website.
>
> ---
>
> **Q:** I am not sure why the "oracle" is called an "oracle" rather than a rule-based baseline
>
> **A:** Thanks for the suggestion, we will revise the paper to explain this. We call it an “oracle” since it assumes knowledge of an explicit hierarchical form, where the higher levels are assumed to be known or explicitly provided in a rule-based form. This is in contrast to all our learned policies which do not assume any known components or a hierarchical structure.
>
> ---
>
> **Q:** I am not convinced why the approach would work better than learning a policy separately for each entity.
>
> **A:** In order to clarify the suggested approach, how would an approach with one policy per entity extrapolate to tasks with more entities than seen during training? This would potentially require an explicit hierarchical or rule-based structure on top of a single object policy. Our setup trains a policy end-to-end on a certain number of objects, and evaluates the policy zero-shot to a varying number of objects.
>
> ---
>
> **Q:** The proposed architecture can outperform MLP because it simply has more learning capacity. The authors should show the number of parameters of the models in the comparisons.
>
> **A:** We show here the number of parameters for the actor and critic when doing RL on 3-Push, and have also added these numbers to the appendix. Note that the **Deep set has fewer parameters, yet outperforms MLP** quite considerably across our evaluations, suggesting that invariance, not capacity, is the difference maker. The Self Attention policy contains more parameters, though this is difficult to control since transformer-style networks are rarely trained with such a small number of parameters.
>
> | Model Class | # actor parameters | # critic parameters |
> | -------------- | ---------------------- | ---------------------- |
> | MLP | 150,020 | 150,273 |
> | Deep Set | 140,292 |140,545 |
> | Self Attention | 800,260 | 800,513 |
> | Graph Conv | 271,876 | 272,129 |

---

> > ### Author Response · Authors · 2022-11-18
> > **Response to Reviewer KcU5 (2/2)**
> >
> > **Q:** When training the MLP, do you permute the input entity subgoals in each episode or do you always feed the subgoals in the same order?
> >
> > **A:** We actually initially tried permuting the entity order for the MLP to encourage invariance, but this failed completely (the policy failed to train under RL at all) so we kept the entities in a fixed order instead. We conjecture that current RL algorithms, like DDPG+HER, are not capable of learning with strong permutation based augmentations. We will update the paper with these results.
> >
> > ---
> >
> > **Q:** The conclusion says "These policy architectures decompose goal-conditioned tasks into their constituent entities and subgoals" but this seems like a false claim, as the goal specification already consists of entity subgoals stitched together.
> >
> > **A:** We have updated the paper to clarify the wording here. We do not mean to suggest that these policies are “decomposing” entity representations, since we are operating on a low-dimensional state representation. Rather, they decompose tasks into per-entity subtasks, as evidenced by their ability to extrapolate to a varying number of entities/subtasks.
> >
> > ---

---

### Official Review · Reviewer_dhRF · 2022-10-22

**Confidence:** 3
**Correctness:** 3
**Technical Novelty And Significance:** 3
**Empirical Novelty And Significance:** 3
**Recommendation:** 8

**Clarity, Quality, Novelty And Reproducibility:**

- As mentioned, the paper is well-written and easy to follow. The writing quality is high.
- As far as I know, the paper has good novelty in terms of its EFMDP framework, but I’m not certain if similar ideas exist in multi-agent RL literature, maybe other reviewers could also verify this.
- The paper should be easy to produce since it provides detailed experiment settings, and the authors have also promised to release the code upon publication.

**Strength And Weaknesses:**

**Strength:**

- The paper presents a neat and intuitive idea to factorize the state and goal configurations across entities, which is beneficial to generalization and scaling.
- Experiments are extensive, where multiple permutation-invariant architectures are studied in various different settings, e.g., zero-shot extrapolation and zero-shot stitching. The results demonstrate the effectiveness of the proposed method and empirically show how different architectures compare for these tasks.
- The paper is well structured with clear problem definitions, and how different architectures fit in the big picture. The method is well-motivated and easy to follow.

**Weakness:**

- The method is only evaluated on robot manipulation, it would be nice to also evaluate on other settings such as multi-agent systems, and strategic game playing, which would make the experiments stronger.
- While the paper shows different permutation-invariant architectures such as Deep Sets, Graph Convolution, and Self Attention can fit into EFMDP, the framework itself doesn’t provide any insights or guidelines as to which one is better in certain settings, which makes the theoretical analysis less useful.

**Summary Of The Paper:**

This paper presents a framework called Entity-Factorized Markov Decision Process (EFMDP), where the task is factorized across different entities, and the state and goal configurations of the entities are permutation-invariant. Based on this framework, the paper studies various permutation-invariant architecture designs, including Deep Sets, Graph Convolution, and Self Attention. Experiments on object pushing, switching, and stacking demonstrate that the method with permutation-invariant architectures learns faster than MLPs when there are multiple objects in the scene, and can do zero-shot extrapolation to an unseen number of objects.

**Summary Of The Review:**

Overall, I think this is a solid work with good novelty and sufficient experiments. I believe the proposed EFMDP setting could be useful for many other problems as well. Therefore, I vote to accept this paper.


--- After rebuttal ---

I'm happy with the authors' response and decide to keep my score.

---

> ### Author Response · Authors · 2022-11-18
> **Response to Reviewer dhRF**
>
> Thank you for recognizing the novelty of the paper and for providing constructive feedback! We answer your specific questions below.
>
> **Q:** The method is only evaluated on robot manipulation, it would be nice to also evaluate on other settings such as multi-agent systems
>
> **A:** Thank you for the suggestion. We are working on applying the EFMDP framework to study differences between architectures on multi-agent environments, in particular in the MultiAgent Particle Environment [1]. We will strive to include these in the camera ready version. At the same time, we want to highlight that **we studied multiple tasks** in the broad family of robot manipulation (including stitching, pushing, and stacking). We believe the set of experiments are sufficiently broad to illustrate the power of structure policy architectures in problems that can be modeled as EFMDPs.
>
> [1] Lowe et al. Multi-Agent Actor-Critic for Mixed Cooperative-Competitive Environments.
>
> ---
>
> **Q:** While the paper shows different permutation-invariant architectures such as Deep Sets, Graph Convolution, and Self Attention can fit into EFMDP, the framework itself doesn’t provide any insights or guidelines as to which one is better in certain settings
>
> **A:** This is a good point, and we have updated the conclusion section to reflect this. Our primary contribution in this work is the EFMDP formulation, which can model several problems including multi-object manipulation, and to show that structured policies are particularly well suited for learning in EFMDPs. Which structured policy performs the best might depend on the specifics of the problem. Nevertheless, it is clear that structured policies substantially outperform general-purpose architectures like MLPs.

---

### Official Review · Reviewer_HR7M · 2022-10-24

**Confidence:** 3
**Correctness:** 3
**Technical Novelty And Significance:** 3
**Empirical Novelty And Significance:** 3
**Recommendation:** 6

**Clarity, Quality, Novelty And Reproducibility:**

The paper is largely clear and easy to follow. The proposed framework is novel to the best of my understanding. Experiment details are provided for reproducibility.

**Strength And Weaknesses:**

Strengths

- The paper provides a formal study of understanding compositional tasks, which can be useful for several papers that target this domain, for developing future algorithms

- Understanding deep RL policy architectures is generally not very well-studied, and this paper proposes a formal study of this, which can be useful for to several works in RL.

- The experimental evaluations are very detailed and show the emergence of properties like extrapolation to out-of-domain targets, and stitching of skills for compositional generalization.

Weaknesses

- The limitations of the proposed framework are unclear. It will be good to be precise, clear, and transparent about exactly which king of control problems can be cast in this framework, and exactly when will the benefits be observed empirically, compared to casting the problem in the usual MDP framework,

- No external baselines are compared against. In my understadning, a one-on-one comaprison with *any* RL algorithm that is goal-conditioned is possible, and should be performed so that the it is clear how the algorithms developed in this framework comapre to existing goal-conditioned RL algorithms under the MDP framework.

**Summary Of The Paper:**

This paper proposes a framework for compositional structure modeling in control tasks, that is a modification of the typical Markov Decision Process (MDP). The framework, called Entity-Factored Markov Decision Process provides several insights into developing robot control policy architectures for compositional generalization. Results on simulated robot manipulation tasks show extrapolation to different number of test entities, and stitching of skills in novel ways.

**Summary Of The Review:**

I think the paper has interesting insights and opens up possibilities for future works to build upon, and I am leaning towards acceptance. Please look at the details of strengths/weaknesses above.

------ AFTER AUTHOR RESPONSES ----

The authors have updated the limitations section of the paper to include more details, so the current paper is better scoped in terms of its contributions. My second point regarding comparisons to baselines is not adequately addressed and would likely require significant additional work: I was referring to comparisons with other goal-conditioned RL approaches (not just RL + HER) for example approaches described in this paper [A]. Currently, the baselines are not "external" goal-conditioned RL approaches in the sense that they do not correspond directly to a prior paper.

In addition, after going through the other reviews, some concerns regarding proper ablations to tease out the benefits in performance are unaddressed, as overall performance is not indicative of whether confounding factors were at play.

Based on these, I am not able to strongly recommend accept, but would keep my rating of weak accept with lower confidence as I think the paper is interesting for future work to build on

[A] Liu, Minghuan, Menghui Zhu, and Weinan Zhang. "Goal-conditioned reinforcement learning: Problems and solutions." arXiv preprint arXiv:2201.08299 (2022).

---

> ### Author Response · Authors · 2022-11-18
> **Response to Reviewer HR7M**
>
> Thank you for the valuable review and constructive feedback! Please find specific responses to your questions below.
>
> **Q:** The limitations of the proposed framework are unclear
>
> **A:** We have expanded the discussion of limitations (Sec 5), e.g. situations where the framework is and isn’t applicable, and the extent to which the framework can explain performance differences between architectures.
>
> **Q:** No external baselines are compared against. In my understanding, a one-on-one comparison with any RL algorithm that is goal-conditioned is possible
>
> **A:** We believe there may be a misunderstanding here. To clarify, all architectures (including the MLP baseline) are trained with goal-conditioned RL -- specifically DDPG + Hindsight Experience Replay. We also show some results for PPO instead of DDPG in Appendix D.3, which provide the same conclusion -- structured architectures outperform MLPs. We are happy to consider any other specific baseline you have in mind and run them.

---

### Official Review · Reviewer_Et3S · 2022-10-25

**Confidence:** 3
**Correctness:** 4
**Technical Novelty And Significance:** 2
**Empirical Novelty And Significance:** 2
**Recommendation:** 5

**Clarity, Quality, Novelty And Reproducibility:**

The paper is well-written and easy to follow. The results can be reproduced with reasonable effort.


**Strength And Weaknesses:**

### Strength
- The experiments are well designed, and the results support the author's statement well.

### Weakness
- Major:
  - As the authors mentioned also, the current model takes vector state as input. However, a model that can handle visual input is more desired. A related work, SMORL[1], has shown promising results given visual input. Similar results can also be obtained from there.


-------------After rebuttal-----------------------

Thanks for the author's responses. I decide to increase my score to 5.
1)I still think the proposed work shares some common motivations with SMORL, such as the reduced size of effective state space. The authors did investigate more architecture than the SMORL work. However, the experiment's only finding is that utilizing the factorized entity-based structure improves the performance, which does not have much novelty. Similar findings have been demonstrated in prior work like SMORL. 2)Though given the ground-truth entity states and goal state as input, the proposed work is solving a much simpler task than SMORL, I'm convinced that the OOD experiments are meaningful, and I'd like to increase my score to 5. However, I am leaning toward rejection based on the current experimental results.


[1]: Andrii Zadaianchuk, Maximilian Seitzer, and Georg Martius. Self-supervised visual reinforcement learning with object-centric representations.

**Summary Of The Paper:**

This paper introduced the Entity-Factored Markov Decision Process (EFMDP) for modeling the entity-based compositional structure in controlling tasks. The authors studied several structured policy architectures that can utilize the factorized discrete entities on a suite of manipulation tasks. Experimental results showed that structured policy architectures have faster learning speed in general when compared to MLP policy. Robust extrapolation and OOD generalization at the skill-level can also be observed.

**Summary Of The Review:**

This paper introduced the Entity-Factored Markov Decision Process (EFMDP) for modeling the entity-based compositional structure in controlling tasks. Experiments showed that structured policy outperforms the MLP policy. However, similar results have been obtained from related work already.

---

> ### Author Response · Authors · 2022-11-18
> **Response to Reviewer Et3S**
>
> **Q:** Relationship to SMORL
>
> **A:** Thank you for the reference to SMORL, we have cited and discussed it in the revised draft. While SMORL does utilize an attention based policy, we note that our work differs substantially from SMORL, both in terms of motivation and empirical results.
>
> Firstly, our work introduces the EFMDP framework to explain **why** structured policies perform well on multi-entity control tasks. This framework helps to explore the symmetries and geometric properties inherent in multi-object manipulation. In contrast to prior work, we systematically train and evaluate **several** structured or entity-centric architectures like Deep Sets, Self Attention, and GNNs.
>
> Secondly, we are not aware of any work (including SMORL) that investigates zero-shot extrapolation and stitching (Sec 3.2 and 3.3). Indeed, SMORL studies generalization for only one task, by evaluating a policy trained in a 2-object environment on a single object environment. In contrast, we study zero-shot extrapolation of policies to fewer as well as larger number of objects on three different tasks (Fig. 6), in addition to stitching multiple skills (Fig 7).
>
> Finally, we believe that our work is distinct from, but complementary to works that combine RL and object-centric representation learning. There have been major advances in computer vision literature in the context of object detection and segmentation, and object-centric features can constitute the state space of EFMDPs, and can be used as input to any of the structured policies we study.

---

### Decision · Program_Chairs · 2023-01-20

**Decision:**

Reject

**Justification For Why Not Higher Score:**

In their response, the authors mention that the contribution of the paper is in showing **why** permutation-invariant policies work well for multi-entity control tasks, however the reviewers find that the results are largely empirical, with relatively little analysis or insight into the results.

**Justification For Why Not Lower Score:**

N/A

**Metareview: Summary, Strengths And Weaknesses:**

The paper considers the problem of learning goal-conditioned policies in settings where the number of entities (objects) and goal compositions vary. The paper proposes the Entity-Factorized Markov Decision Process (EFMDP) framework, which takes advantage of the symmetric nature of entity-centric goal specifications. Using this framework, the paper explores different  architectures that are invariant to the permutation of the inputs (e.g., Self Attention, Graph Convolution, and Deep Sets) and are able to exploit the entity factorization. Experiments on simulated robot manipulation tasks demonstrate that the framework is capable of zero-shot extrapolation to different numbers of entities, and that the permutation-invariant architectures outperform standard MLP baselines in terms of learning efficiency.

The paper received four reviews that exhibit a notable amount of variability in their overall assessment of the paper. That said, the reviewers generally agree that the paper is well written, that it addresses an interesting problem, and that the experimental results provide a empirical demonstration of the effectiveness of the EFMDP framework.

The reviewers disagreed regarding the novelty of the proposed framework and the significance of the contributions. Two of the reviewers (KcU5 and Et3S) comment that the use of object-centric representations as a means of achieving structured policies is not new. The theoretical focus is on the role of permutation invariance and while the authors claim that the experiments demonstrate why permutation invariant policies are effective for multi-entity control tasks, some reviewers initially commented that this is not supported by the experiments, which are instead about generalization to novel compositions and largely empirical, resulting in a mismatch between the theoretical discussion and the experimental evaluation.

In light of the disparity in the reviews, the AC held a virtual meeting with three of the reviewers (Et3S, dhRF, and HR7M; Reviewer KcU5 was traveling and not available to meet, but did follow up with the AC and reviewers), during which the strengths and weaknesses of the paper were discussed at-length. While the reviewers were not able to come to a full consensus, they generally agreed that the paper provides more empirical evidence compared to recent work (notably SMORL), but that beyond this empirical evidence in terms of success rate, the analysis of the different architectures and insight into their performance is lacking. A revised version of the paper that better aligned theory with the experimental evaluation, including more detailed analysis and insights into the results would provide a valuable contribution.

**Summary Of Ac-Reviewer Meeting:**

A virtual (Zoom-based) meeting was held that involved Reviewers Et3S, dhRF, and HR7M (Reviewer KcU5 was traveling and not available to meet, but did follow up with the AC and reviewers). The following bulleted list summarizes this discussion:


* Et3S: Main concern is that reviewer has learned similar conclusions from existing work (SMORL, ICLR 2021). Experimental results are not surprising
    ** Input to SMORL is images not state as in this paper (i.e., a harder problem)
* dhRF: Knowing more about SMORL, reviewer feels that the algorithmic contributions are less significant.
* Et3S: increased score due to clarity of zero-shot extrapolation and stitching
* dhRF: Paper provides more experimental evaluations relative to SMORL. The question is whether the empirical contributions are sufficient.
* HR7M: Experiments in this paper are far more extensive.
* HR7M: Because paper doesn't focus on representation learning (which S), feels that the conclusions are more robust than with SMORL.
* Et3S: Doesn't feel that the experimental contributions are sufficient for ICLR
* Et3S: Feels that factorization is straightforward. Not aware of a specific paper that does this, but feels that there is existing work that shows the advantages of factorization.
    ---> Consistent with REt3S at a high level
* Et3S: Performance of different architectures is not consistent across experiments. If paper could provide more insights into architecture design, it would provide a valuable contribution.
* Et3S/dhRF: Authors effectively tried different architectures and report the results, but don't provide any analysis or insight into their performance.
* HR7M: Paper only provide success rates and nothing else for analysis, which is what reviewer meant by the lack of ablations that would better explain the difference in performance. Reviewer doesn't feel that this is a significant reason not to accept the paper
* Where in the paper does it show "why structured policies perform well..."?
    ** dhRF believes that this is a reference to Proposition 1 and asks whether this is trivial
        ** HR7M had to look at the proof to see why this is the case. Doesn't feel that this is trivial.
    ** Et3S believes that it is also because they can reduce the goal space.
* dhRF: May lower score. Thinks that they were overly positive before.
* Et3S: Wants to keep score
* HR7M: Strengths outweigh the weaknesses.